

# Development of a prognostic model based on the ceRNA network in Triple-Negative Breast cancer

Yimin Zhu[1], Jiayu Wang[2] and Binghe Xu[2]

[1] Medical Oncology Department, Chinese People's Liberation Army General Hospital, Beijing, China
[2] National Cancer Center/National Clinical Research Center for Cancer/Cancer Hospital, Chinese Academy of Medical Science and Peking Union Medicao College, Beijing, China

## ABSTRACT

**Background.** Triple-negative breast cancer (TNBC) is an aggressive subtype with a poor prognosis. Although circular RNAs (circRNAs) have been implicated in cancer progression, their roles in TNBC remain poorly understood. In this study, we aimed to develop a prognostic model for TNBC by constructing a competing endogenous RNA (ceRNA) network. This network integrates circRNAs, long noncoding RNAs (lncR-NAs), microRNAs (miRNAs), and messenger RNAs (mRNAs) to identify potential biomarkers and therapeutic targets for improving clinical outcomes.

**Methods.** Differentially expressed circRNAs, lncRNAs, and mRNAs were identified from GEO datasets (144 samples: 94 TNBC and 50 normal tissues). A ceRNA network was constructed, and key genes were validated using The Cancer Genome Atlas (TCGA) dataset (115 TNBC and 113 para-cancer tissues). Multivariate Cox regression analysis was performed to develop a prognostic model, and Gene Set Enrichment Analysis (GSEA) was performed to identify associated pathways.

**Results.** Nine genes (*SH3BGRL2*, *CA12*, *LRP8*, *NAV3*, *GFRA1*, *DCDC2*, *CDC7*, *ABAT*, *NPTX1*) were identified as key factors in the prognostic model, which demonstrated an area under the curve (AUC) of 0.90. Patients classified as high-risk patients exhibited significantly shorter overall survival (median OS: 8.12 years *vs*. 9.51 years, $P < 0.01$). The mitogen-activated protein kinase (MAPK) signaling pathway was identified as a key regulatory pathway, with circRNAs (hsa_circ_0005455, hsa_circ_000632, hsa_circ_0001666, and hsa_circ_0000069) regulating *CA12*, *GFRA1*, and *NPTX1* expression.

**Conclusion.** This study developed a novel prognostic model based on a ceRNA network analysis, highlighting the critical role of circRNAs and the MAPK signaling pathway in TNBC progression. These findings offer valuable insights into potential biomarkers for TNBC prognosis and reveal promising therapeutic targets for improving patient outcomes.

## INTRODUCTION

Breast cancer is the primary cause of cancer-associated death in women worldwide. According to recent global cancer statistics, breast cancer accounted for 2.3 million

Corresponding author
Binghe Xu, binghexu2021@163.com

new cases and 685,000 deaths in 2023 (*Sedeta, Jobre & Avezbakiyev, 2023*). The incidence among diagnosed young women aged 15 to 39 years has shown a concerning upward trend (*Yuan et al., 2024*). Genetic predisposition, particularly mutations in Breast Cancer type 1 susceptibility protein (*BRCA1*)/Breast Cancer type 2 susceptibility protein (*BRCA2*), significantly increases breast cancer risk (*Pal, Das & Pandey, 2024*). Women with a family history or hereditary *BRCA1*/*BRCA2* mutations have a notably higher incidence of the disease (*Miller et al., 2020*). Breast cancer encompasses distinct molecular subtypes with unique characteristics and treatment approaches. The current classification recognizes at least five subtypes: triple-negative breast cancer (TNBC), luminal B(LB), luminal A (LA), Her-2 enriched, and normal breast-like cancer (*Harbeck & Gnant, 2017*). TNBC was characterized by negative expression of estrogen receptor (ER), progesterone receptor (PR), and human epidermal growth factor receptor 2 (HER2), exhibiting aggressive biological behavior and poor prognostic (*Waks & Winer, 2019*). It accounts for approximately 15–20% of all breast cancers and is more prevalent among younger women, African American women, and those with *BRCA1* mutations. Multiple signaling pathways are dysregulated in TNBC, including PI3K/AKT/mTOR, MAPK/ERK, and JAK/STAT pathways, which drive tumor proliferation, invasion, and metastasis (*Xiong, Wu & Yu, 2024*). Current research focuses on identifying targeted therapies and tailoring treatment based on tumor-specific biomarkers (*Wang et al., 2021*; *Zhou et al., 2022*).

Only about 3% of the genome was involved in protein encoding, while the remaining 97% could be transcribed into various RNA species known as noncoding RNA (ncRNAs) (*Diederichs et al., 2016*). The function of ncRNAs was mainly associated with phenotypic regulation. However, recent studies highlighted their significant role as a biomarker for cancers (*Sur et al., 2020*; *Tian et al., 2020*). TNBC involves complex molecular mechanisms driven by various RNA molecules that influence its progression. Three key types of RNA play crucial roles: messenger RNA (mRNA), long noncoding RNA (lncRNA), and circular RNA (circRNA) (*Xia et al., 2021*). mRNAs, which carry genetic instructions for protein synthesis, can promote or suppress tumor development in TNBC. These molecules regulate essential cellular processes, including growth, programmed cell death, invasion, and spread to other tissues (*Anilkumar et al., 2023*). lncRNAs are non-protein-coding RNA molecules that play critical roles in gene regulation, encompassing chromatin remodeling, transcriptional regulation, and post-transcriptional control (*Qian, Shi & Luo, 2020*). In TNBC, lncRNAs modulate tumorigenesis by regulating genes involved in cell proliferation, apoptosis, as well as metastasis (*Zhang, Guan & Tang, 2021*). CircRNAs represent a distinctive class of noncoding RNAs characterized by covalently closed loops, rendering them resistant to exonuclease degradation. These molecules can function as molecular sponges for microRNAs (miRNAs), thereby regulating gene expression by competing for miRNA binding sites, and can also modulate gene transcription through interactions with RNA-binding protein (*Xu et al., 2024*). The dysregulation of circRNAs among TNBC contributes to cancer progression, metastasis, and chemotherapy resistance (*Li et al., 2024*). The role of circRNAs in breast cancer development and progression has emerged as a significant focus of recent years (*Bian, 2019*; *Li et al., 2020b*). Competing endogenous RNA (ceRNA) represents a complex mechanism of gene regulation in which various RNA molecules

contend for binding to the same miRNAs. This regulatory network comprises multiple RNA types, including mRNAs, lncRNAs, and circRNAs, which harbor miRNA Response Elements (MREs), enabling them to contend for miRNA binding (*Qattan et al., 2024*). Accumulating evidence demonstrates that circRNAs function as ceRNA molecules by acting as miRNA sponges, thereby modulating gene expression through sequestration of specific miRNAs in TNBC (*Jiang & Cheng, 2020*; *Li et al., 2020a*; *Qattan, 2024*). These findings have established circRNAs as critical regulatory elements in TNBC pathogenesis and highlighted their potential as novel therapeutic targets.

This study investigates the prognostic potential of circRNA-mediated ceRNA networks in TNBC. Through comprehensive bioinformatic analysis, we developed a prognostic model incorporating nine key genes: *SH3BGRL2*, *CA12*, *LRP8*, *NAV3*, *GFRA1*, *DCDC2*, *CDC7*, *ABAT*, and *NPTX1*. Our analysis identified four differentially expressed circular RNAs (hsa_circ_0005455, hsa_circ_000632, hsa_circ_0001666, and hsa_circ_0000069) that regulate these genes through ceRNA network interactions. Further analysis has suggested the involvement of the MAPK signaling pathway in this regulatory network. These findings provide a foundation for understanding circRNA-based prognostic markers and identifying potential therapeutic targets in TNBC.

## METHODS

### Collections of data from Gene Expression Omnibus and The Cancer Genome Atlas

The eligible GEO should conform to the following criteria: (1) studies with TNBC and normal tissues; (2) the information of platforms and studies were qualified for analysis; (3) the dataset type was microarray.

We retrieved five associated microarray datasets from the Gene Expression Omnibus (https://www.ncbi.nlm.nih.gov/geo/). There were one circRNA expression dataset (GSE101123), two lncRNA expression datasets (GSE64790 and GSE115275), and two mRNA expression datasets (GSE38959 and GSE53752). One hundred forty-four samples were identified, including 94 TNBC and 50 para-cancer tissue samples.

To further analyze the impact of differentially expressed genes (DEGs) on TNBC patients, RNA sequencing expression data as well as clinical information for mRNA, miRNA, and lncRNA were downloaded from The Cancer Genome Atlas (TCGA) database for 115 TNBC cases, 989 non-TNBC cases, and 113 paracancerous tissues (https://www.cancer.gov/ccg/research/genome-sequencing/tcga).

The selection criteria for clinical information of TNBC patients in the TCGA cohort included: (1) histologically confirmed TNBC with negative ER, PR, and HER2 status; (2) complete clinical follow-up data ≥ 90 days; (3) clinical information such as age, tumor stage, treatment history, and survival outcomes (overall survival time and status).

### Analysis of the differently expressed lncRNA, circ-RNAs, and mRNAs by microarrays

The differential expression analysis was conducted using multiple microarray datasets from the GEO database, where circRNA analysis utilized dataset GSE101123 with annotations

from the GPL19978 platform (Agilent-069978 Arraystar Human CircRNA microarray V1). LncRNA expression was analyzed using two independent datasets: GSE64790 (annotated using GPL19612; Agilent-062978 Human lncRNA v4 Microarray) and GSE115275 (annotated using GPL21827; Agilent-079487 Human lncRNA v4 Microarray), while mRNA expression profiling integrated data from GSE38959 and GSE53752, annotated using GPL4133 (Agilent-014850 Whole Human Genome Microarray 4x44K G4112F) and GPL7264 (Agilent-012097 Human 1A Microarray G4110B) platforms respectively. To mitigate batch effects and technical variations arising from different platforms and experimental time points, we implemented batch correction using the Sva package in R (version 4.3.0; *R Core Team, 2023*), where the ComBat function was applied with default parameters to harmonize the data across different batches while preserving biological variation. Differential expression analysis was conducted with the limma package in R-Bioconductor, including log2 transformation of expression values, linear model fitting using the lmFit function, and empirical Bayes statistics using the eBayes function with default parameters. The |log FC| > 1 and FDR-*P* values < 0.05 were considered as the available threshold for identifying differential expression genes.

## Construction of the competing endogenous RNA (circRNA/miRNA/mRNA/LncRNA)

We systematically predicted RNA interactions to construct a ceRNA network based on the differential expression results from GEO datasets. The DEmRNA-miRNA interaction pairs were predicted through the integrated analysis of TargetScan (version 7.2, https://www.targetscan.org/vert_80/) and miRDB (version 5.0, https://mirdb.org/), with the intersection of both databases' predictions being used to minimize false positives. The DElncRNA-miRNA interactions were identified using miRcode (version 11; http://www.mircode.org/). For DEcircRNA-miRNA interaction pairs, StarBase (Version 2.0; https://rnasysu.com/encori/) was employed to identify interaction pairs. The ceRNA network was visualized using Cytoscape (version 3.6.1) with RNA type-specific color coding (circRNAs: red circles, miRNAs: light blue diamond, mRNAs: green triangle, lncRNAs: Blue square).

## Gene Ontology and Kyoto Encyclopedia of Genes and Genomes analysis of differently expressed mRNA in the ceRNA network

We performed a comprehensive functional enrichment analysis of the differentially expressed mRNAs identified in the ceRNA network using Gene Ontology (GO) and Kyoto Encyclopedia of Genes and Genomes (KEGG) annotations. The analysis was conducted using the clusterProfiler R package with Bioconductor 3.18. For GO analysis, we utilized the org.Hs.eg.db annotation package with GO database version, examining all three GO categories: Biological Process (BP), Molecular Function (MF), and Cellular Component (CC). We employed the enrichKEGG function with parameters organism = "hsa" and default parameters for KEGG analysis. The results were visualized using the enrichplot package. *P* values < 0.05 were considered the threshold for identifying significant GO and KEGG.

## Screening the co-DEGs both in ceRNA and TCGA

Initial data preprocessing included filtering to retain genes with mean expression values > 0.2. The R-Bioconductor limma package was applied to analyze the expression of genes from TCGA, and the $|\log FC| > 1$ and FDR-$P$ values < 0.05 were considered as the available threshold for identifying differential expression genes between TNBC/non-TNBC cancer tissues and para-cancer tissues. Venn intersection analysis determined the co-DEGs in TCGA and ceRNA.

## Development of a prognostic model

Clinical information and expression data for the co-differentially expressed genes were collected from the TCGA. The clinical data included age, tumor stage, treatment history, and survival outcomes (overall survival time and status). Data preprocessing involved removing samples with missing survival information or a follow-up time of less than 90 days. Multivariate and univariate Cox proportional hazards regression analysis was performed using the R survival and gtsummary package to develop a prognostic index. Patients were stratified into high-risk as well as low-risk groups using the median risk score as the cutoff. Survival analysis was conducted using the survival package and survminer. The survivalROC and PRROC package was employed to examine the predictive accuracy of the prognostic model. Additionally, we developed three distinct prognostic models: Model I (9-gene signature): *SH3BGRL2, CA12, LRP8, NAV3, GFRA1, DCDC2, CDC7, ABAT,* and *NPTX1.* Model II (6-gene signature): *SH3BGRL2, CA12, NAV3, GFRA1, ABAT,* and *NPTX1.* Model III (5-gene signature): *SH3BGRL2, CA12, NAV3, CDC7,* and *ABAT.*

## Gene set enrichment analysis of DEGs associated with a prognostic model

To investigate the underlying biological mechanisms of the prognostic index, we performed Gene Set Enrichment Analysis (GSEA) using GSEA software version 4.0.3. The analysis used the Molecular Signatures Database (MSigDB) version 6.0, specifically the c2.cp.kegg.v6.0.symbols.gmt dataset, which contains 186 curated KEGG pathway gene sets. Gene set size filters ≥ 15. Statistical significance was assessed using normalized enrichment score (NES), nominal $P$-value < 0.05, and false discovery rate (FDR) < 0.25. The results were visualized using enrichment plots showing the running enrichment score and positions of gene set members on the ranked list.

## Statistical analysis

Categorical variables were presented as frequencies and percentages (%). The normality of continuous variables was assessed using the Shapiro–Wilk test. Normally distributed continuous variables were expressed as mean ± standard error (SE), while non-normally distributed continuous variables were presented as median with interquartile range (IQR, 25th–75th percentiles). Between-group comparisons for categorical variables were performed using Pearson's chi-square test or Fisher's exact test. An independent samples $t$-test was used for continuous variables with normally distributed data, while the Mann–Whitney U test was applied for data that were not non-normally distributed. All statistical

tests were two-sided, and $P < 0.05$ was considered statistically significant. All statistical analyses and visualizations were conducted using the R (version 4.3.0).

## RESULTS

### Analysis of differentially expressed mRNA/lncRNA/circRNAs in TNBC compared to normal breast tissue from the GEO database

After downloading the raw data and platform information from the GEO database, two different mRNAs/lncRNAs datasets were normalized in batch. Then, they merged to increase the number of patients and data reliability. Based on the cutoff value with $|\log FC| > 1$, $P$ values $< 0.0$. One hundred forty differentially expressed circRNAs, 493 differentially expressed mRNAs, and 2,998 differently expressed LncRNAs were identified, and the heatmap demonstrated distinct clustering patterns between TNBC and paired non-tumorous tissues based on the expression profiles of circRNAs, mRNAs, and LncRNAs (Fig. 1).

### Construction of CeRNA (CircRNAs/LncRNAs/miRNAs/mRNAs) network

To elucidate the complex regulatory interactions among differentially expressed RNAs in TNBC, we constructed a ceRNA network. The initial prediction analysis using the miRcode database identified significant interactions between 34 differentially expressed lncRNAs and 203 miRNAs (Table S1). Further analysis through the integration of TargetScan and miRDB databases revealed regulatory relationships between differentially expressed 138 mRNAs and 193 miRNAs (Table S2). Using the StarBase database, we further identified additional interactions involving 20 differentially expressed circRNAs and 167 miRNAs (Table S3). After integrating these results and filtering for common interactions, we finalized a ceRNA network comprising 15 circRNAs, 34 lncRNAs, 78 miRNAs, and 107 mRNAs. This interconnected regulatory network was visualized using Cytoscape software to illustrate the complex RNA-RNA interactions in TNBC, while the connection degree of each gene was calculated to demonstrate its contribution in the ceRNA network (Fig. 2 and Table S5).

### GO and KEGG pathway analysis of DEGs in CeRNA

To investigate the biological functions and signaling pathways associated with the DEGs in the ceRNA network, we performed Gene GO term and KEGG pathway analyses. GO analysis revealed significant enrichment in several key biological processes ($P < 0.05$), including negative regulation of protein phosphorylation, negative regulation of phosphorylation, cellular response to alcohol, oxygen level sensing, metal ion response, hypoxia adaptation, and negative regulation of the MAPK cascade (Fig. 3A). KEGG pathway analysis identified four significantly enriched pathways ($P < 0.05$): the MAPK signaling pathway, microRNAs in cancer, transcriptional misregulation in cancer, and the FOXO signaling pathway (Fig. 3B). Notably, the MAPK signaling pathway was the most significantly enriched ($P = 0.005$), suggesting its potential central role in TNBC progression through the ceRNA network regulation.
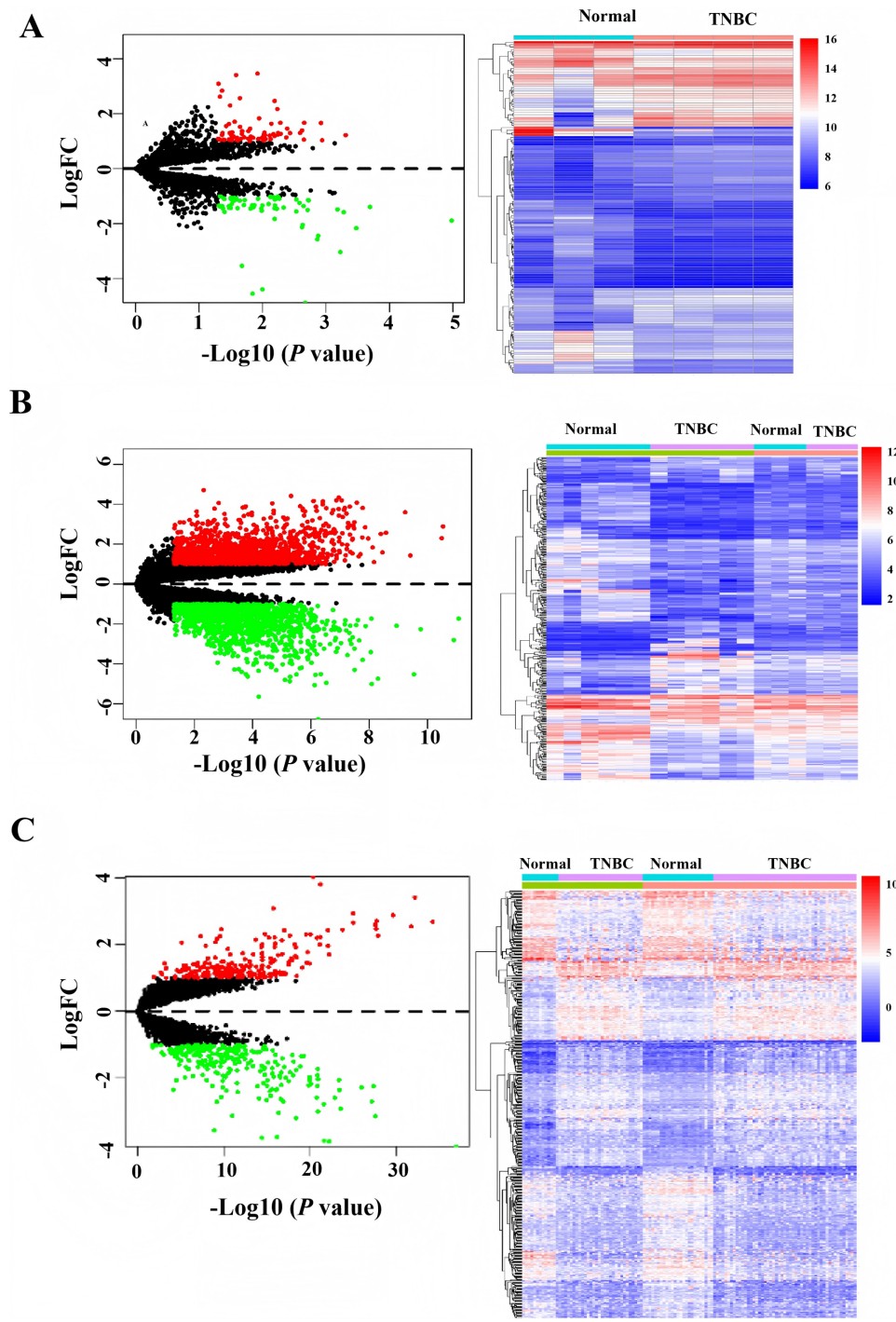

**Figure 1  Heatmap and volcano of differentially expressed genes in GEO database.** (A) Heatmap of differentially expressed circRNAs. (B) Heatmap of differentially expressed LncRNAs. (C) Heatmap of differentially expressed mRNAs.

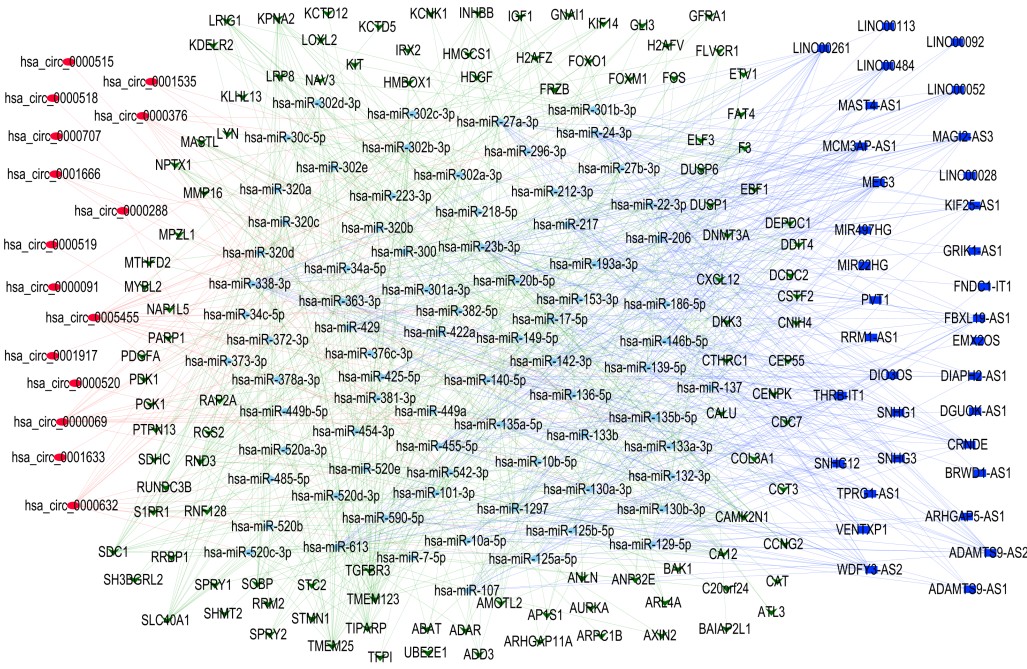

**Figure 2** **Competing endogenous RNA network.** Red circle indicates circRNAs; Blue square indicates lncRNAs; Green triangle indicates mRNAs; Blue diamond indicates miRNAs.

## Identification of co-expressed DEGs in CeRNA and TCGA

To identify TNBC-specific gene signatures, we conducted differential expression analysis comparing TNBC samples with non-TNBC breast cancer tissues and normal tissues using RNA-sequencing data from the TCGA database (Figs. 4A and 4B). Through filtering criteria ($|\log FC| > 1$, FDR-adjusted $P < 0.05$), a total of 1,981 DEGs exclusively expressed in TNBC were identified (Fig. 4C). Subsequently, a Venn diagram analysis was performed to identify genes in our constructed ceRNA network that were differentially expressed in TNBC samples. This intersection analysis revealed nine key genes: *SH3BGRL2*, *CA12*, *LRP8*, *NAV3*, *GFRA1*, *DCDC2*, *CDC7*, *ABAT*, and *NPTX1* (Fig. 4D), suggesting their potential role as TNBC-specific molecular markers.

## Development of a prognostic model utilizing the nine DEGs

One hundred and four TNBC patients with an overall survival time exceeding 90 days were included for subsequent analysis. As demonstrated in Table 1, significant differences were observed in the clinical stage ($P = 0.005$) and chemotherapy history ($P = 0.02$) exhibited between the survival and mortality groups. Furthermore, univariate and multivariate Cox regression analysis suggested that risk score derived from nine DEGs ($P = 0.008$) emerged as a statistically significant independent predictive factor (Table 2). The risk score formula was calculated using the following formula: Risk Score = $[-0.79 \times SH3BGRL2]$ + $[0.40 \times CA12]$ + $[0.16 \times LRP8]$ + $[1.98 \times NAV3]$ + $[0.37 \times GFRA1]$ + $[0.23 \times DCDC2]$ + $[-0.55 \times CDC7]$ + $[-1.18 \times ABAT]$ + $[-0.90 \times NPTX1]$. Patients were stratified into two groups using the median risk score as the threshold value (Fig. 5A).

A

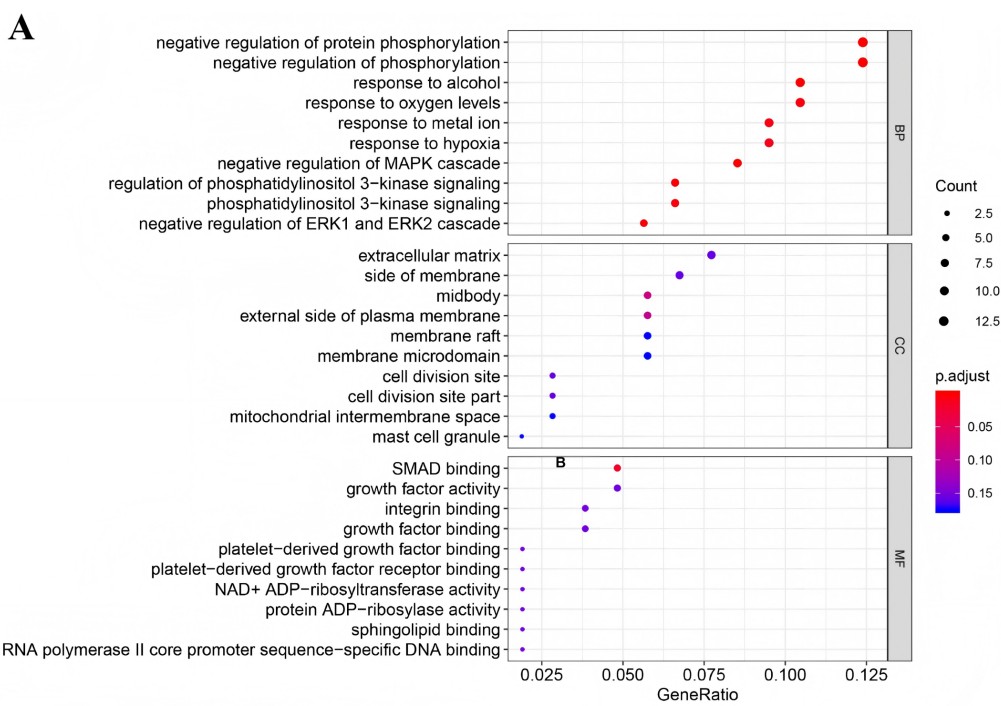

B

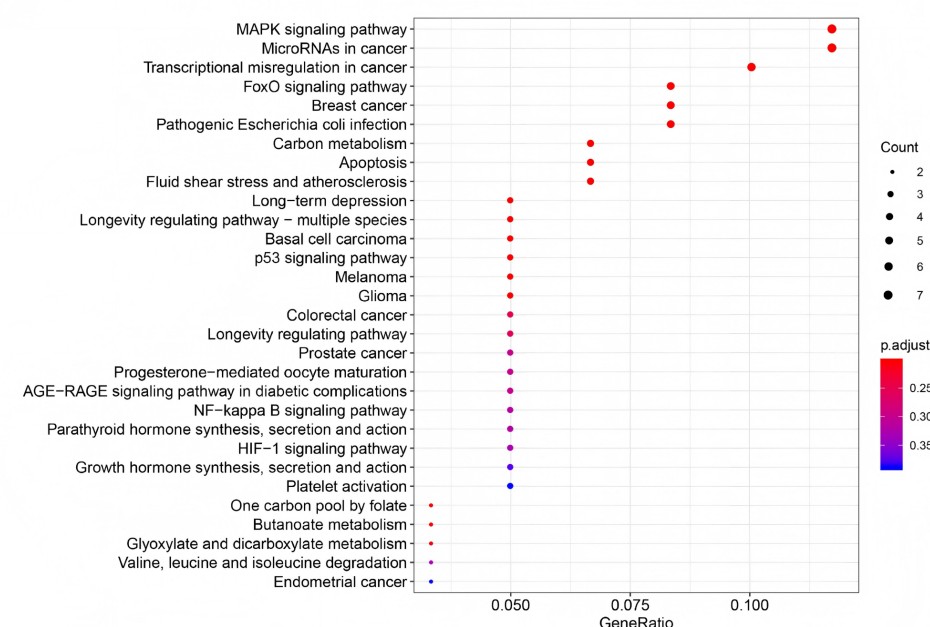

**Figure 3** **Functional analysis of mRNAs in the ceRNA network.** (A) Gene Ontology enrichment analysis of mRNAs. (B) KEGG pathways of mRNAs.

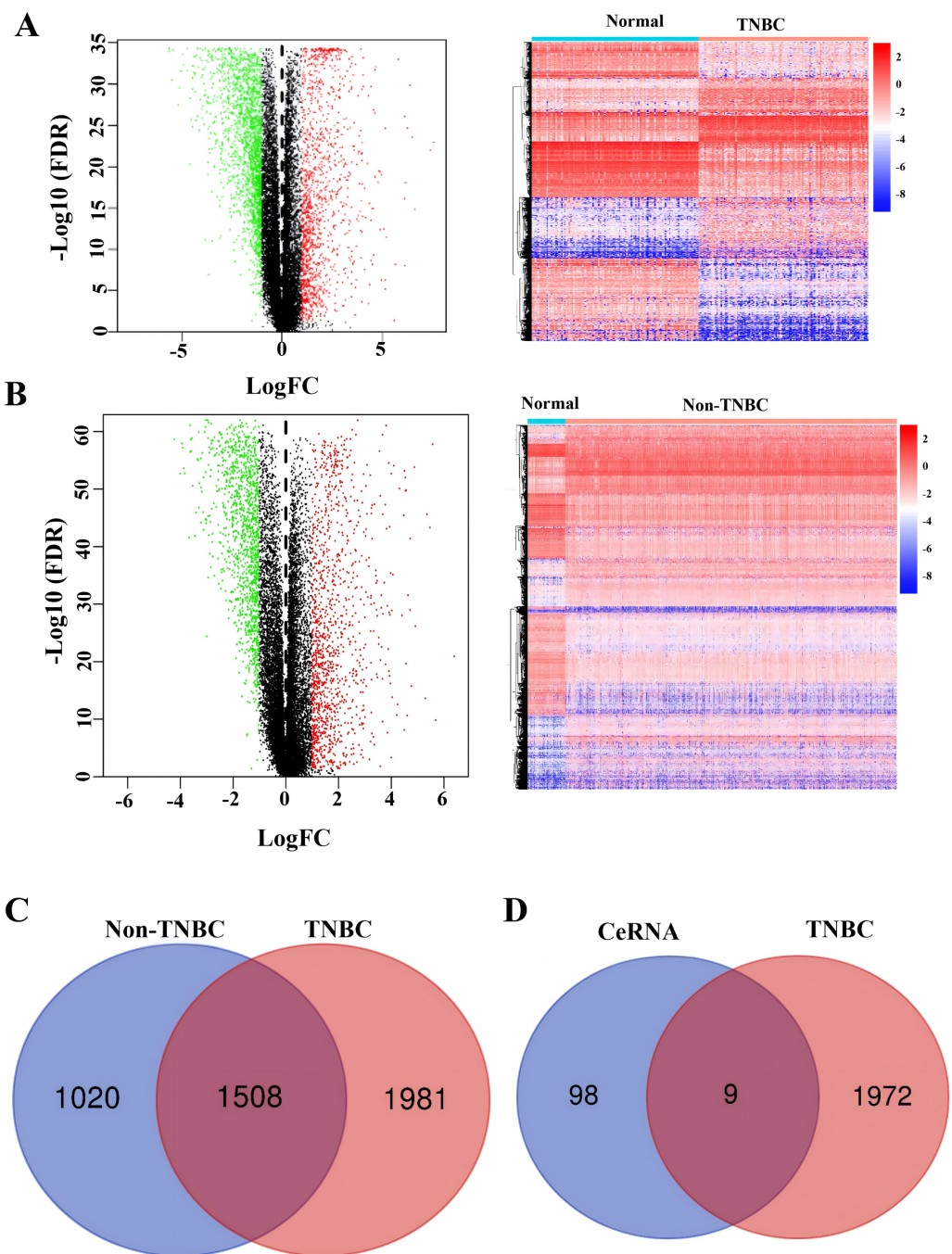

**Figure 4 Identification of co-expressed DEGs in CeRNA and TCGA.** (A) Volcano plot and heatmap of differently expressed genes in TNBC tissue. (B) Volcano plot and heatmap of differently expressed genes in non-TNBC tissue. (C) The Venn analysis of DEGs between TNBC and Non-TNBC tissue. (D) The Venn analysis of DEGs between CeRNA network and DEGs exclusively expressed in TNBC from TCGA dataset.

**Table 1  The basic clinical characteristic of TNBC patients.**

|  | Survival<br>N = 87 | Death<br>N = 17 | P value |
|---|---|---|---|
| Age | 53.0<br>[46.0–61.0] | 50.0<br>[48.0–60.0] | 0.666 |
| Stage |  |  | 0.005 |
| Stage I | 18 (20.7%) | 1 (5.88%) |  |
| Stage II | 54 (62.1%) | 7 (41.2%) |  |
| Stage III | 13 (14.9%) | 6 (35.3%) |  |
| Stage IV | 0 (0.00%) | 2 (11.8%) |  |
| Unknown | 2 (2.30%) | 1 (5.88%) |  |
| Chemotherapy |  |  | 0.020 |
| No | 13 (14.9%) | 7 (41.2%) |  |
| Yes | 74 (85.1%) | 10 (58.8%) |  |
| Hormone therapy |  |  | 0.187 |
| No | 84 (96.6%) | 15 (88.2%) |  |
| Yes | 3 (3.45%) | 2 (11.8%) |  |
| Surgery |  |  | 0.387 |
| Lumpectomy | 28 (32.2%) | 4 (23.5%) |  |
| Modified radical mastectomy | 24 (27.6%) | 4 (23.5%) |  |
| Others | 14 (16.1%) | 6 (35.3%) |  |
| Simple mastectomy | 21 (24.1%) | 3 (17.6%) |  |
| Radiation |  |  | 0.321 |
| No | 82 (94.3%) | 15 (88.2%) |  |
| Yes | 5 (5.75%) | 2 (11.8%) |  |
| Additional radiation |  |  | 0.069 |
| No | 86 (98.9%) | 15 (88.2%) |  |
| Yes | 1 (1.15%) | 2 (11.8%) |  |
| Post chemotherapy |  |  | 1.000 |
| No | 73 (83.9%) | 15 (88.2%) |  |
| Yes | 14 (16.1%) | 2 (11.8%) |  |

As shown in Fig. 5B, Kaplan–Meier survival analysis indicated that the low-risk score group exhibited significantly improved prognosis ($P < 0.01$). The area under the curve (AUC) of receiver operating characteristic (ROC) was 0.90, indicating that this prognostic signature demonstrates superior predictive capability for clinical outcomes in TNBC patients compared to alternative models with AUCs of 0.86 and 0.83 (Fig. 6A). Similarly, in the precision-recall (PR) Curve analysis, model I achieved an AUC of 0.46, outperforming Model II and matching the performance of Model III (Fig. S1). Moreover, elevated risk scores corresponded to increased mortality risk (Fig. 6B).

## GSEA of the DEGs based on the prognostic model

To identify the signal pathway associated with the prognostic model, we employed the GSEA method to analyze the expression profile of the nine genes in the model. The results revealed that the high expression of *CA12*, *GFRA1*, and *NPTX1* was significant on the

**Table 2  Cox regression analysis of the variables.**

| Variables | Univariable | | | Multivariable | | |
|---|---|---|---|---|---|---|
| | HR | 95% CI | *P* value | HR | 95% CI | *P* value |
| Age | 1.00 | 0.96–1.03 | 0.8 | | | |
| Stage | | | | | | |
|   Stage I | – | – | | – | – | |
|   Stage II | 1.94 | 0.23–16.3 | 0.5 | 1.79 | 0.21–15.2 | 0.6 |
|   Stage III | 13.7 | 1.63–116 | 0.016 | 8.34 | 0.88–79.0 | 0.065 |
|   Stage IV | 47.6 | 4.05–560 | 0.002 | 85.6 | 1.29–5663 | 0.037 |
|   Unknown | 4.32 | 0.25–73.9 | 0.3 | 1.73 | 0.09–33.5 | 0.7 |
| Chemotherapy | | | | | | |
|   No | – | – | | – | – | |
|   Yes | 0.36 | 0.13–0.97 | 0.043 | 0.42 | 0.12–1.39 | 0.2 |
| Hormone therapy | | | | | | |
|   No | – | – | | | | |
|   Yes | 4.06 | 0.90–18.3 | 0.069 | | | |
| Surgery | | | | | | |
|   Lumpectomy | – | – | | | | |
|   Modified radical mastectomy | 1.25 | 0.31–5.09 | 0.8 | | | |
|   Other | 2.32 | 0.64–8.41 | 0.2 | | | |
|   Simple mastectomy | 0.94 | 0.21–4.23 | 0.9 | | | |
| Radiation | | | | | | |
|   No | – | – | | | | |
|   Yes | 2.31 | 0.52–10.3 | 0.3 | | | |
| Additional radiation | | | | | | |
|   No | – | – | | | | |
|   Yes | 6.61 | 1.40–31.3 | 0.017 | 0.79 | 0.02–31.4 | 0.9 |
| Post chemotherapy | | | | | | |
|   No | – | – | | | | |
|   Yes | 1.55 | 0.34–7.00 | 0.6 | | | |
| Risk score | 1.16 | 1.09–1.24 | <0.001 | 1.11 | 1.03–1.20 | 0.008 |

common MAPK signal pathway (Fig. 7), which consisted of the KEGG analysis of CeRNA (Fig. 3B). In addition, both the GEO and TCGA datasets demonstrated significantly reduced expression of *CA12* ($P < 0.01$), *GFRA1* ($P < 0.01$), and *NPTX1* ($P < 0.01$) in TNBC tissue (Fig. 8).

## Identification of the CircRNAs associated with MAPK signal pathway

Based on the GSEA analysis result, we constructed a ceRNA network to identify circRNAs associated with the MAPK signal pathway (Fig. 9A and Table S4). As illustrated in Fig. 9B, hsa_circ_0005455, hsa_circ_000632, hsa_circ_0001666, and hsa_circ_0000069 emerged as key regulators of *CA12*, *GFRA1*, *NPTX1*, which are critical components of the MAPK signal pathway. Specifically, hsa_circ_0005455 regulates NPTX1 through hsa-miR-135a/b-5p, hsa-miR-130a/b-3p, hsa-miR-454-3p, and hsa-miR-301a/b-3p.

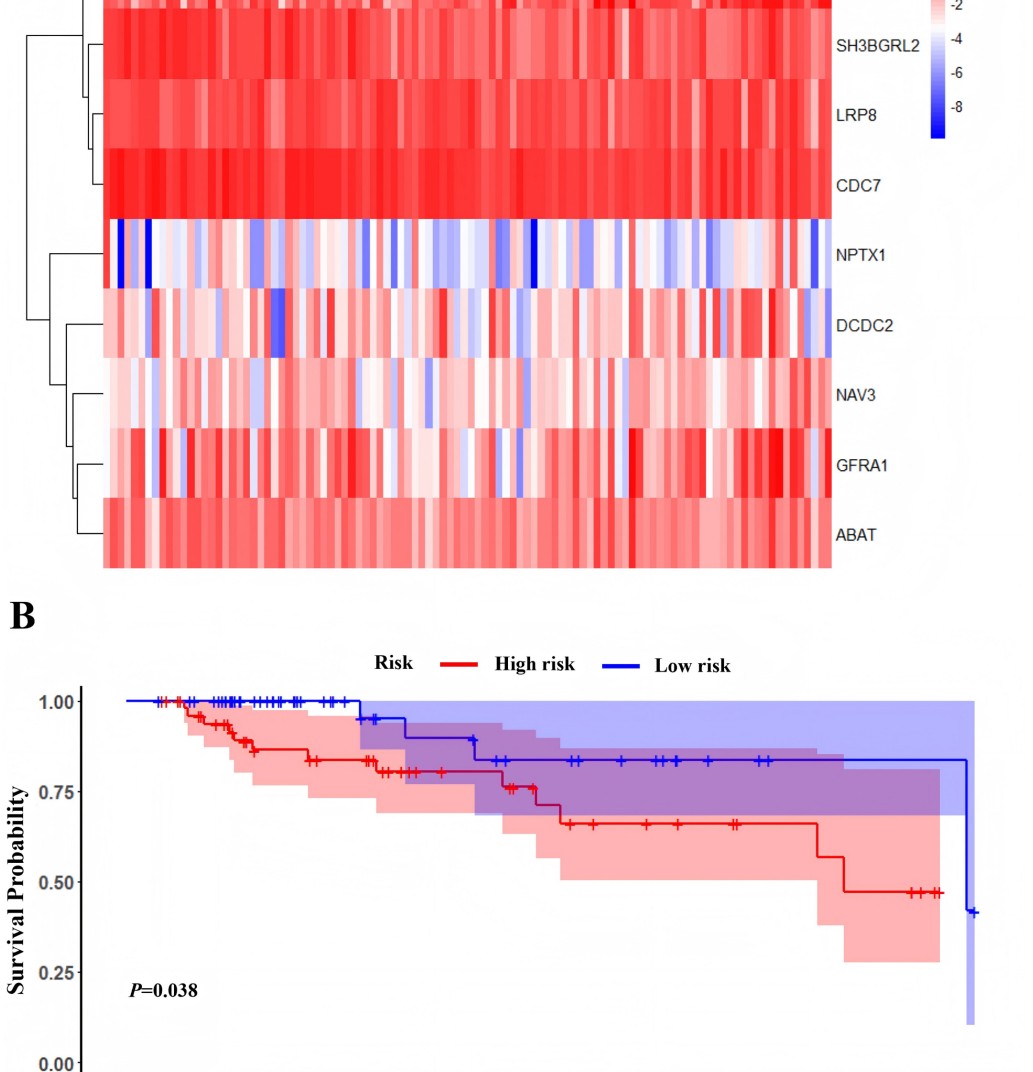

**Figure 5** **Development of prognostic Model.** (A) Heatmap of expression profile of included genes. (B) Survival status of patients in different groups.

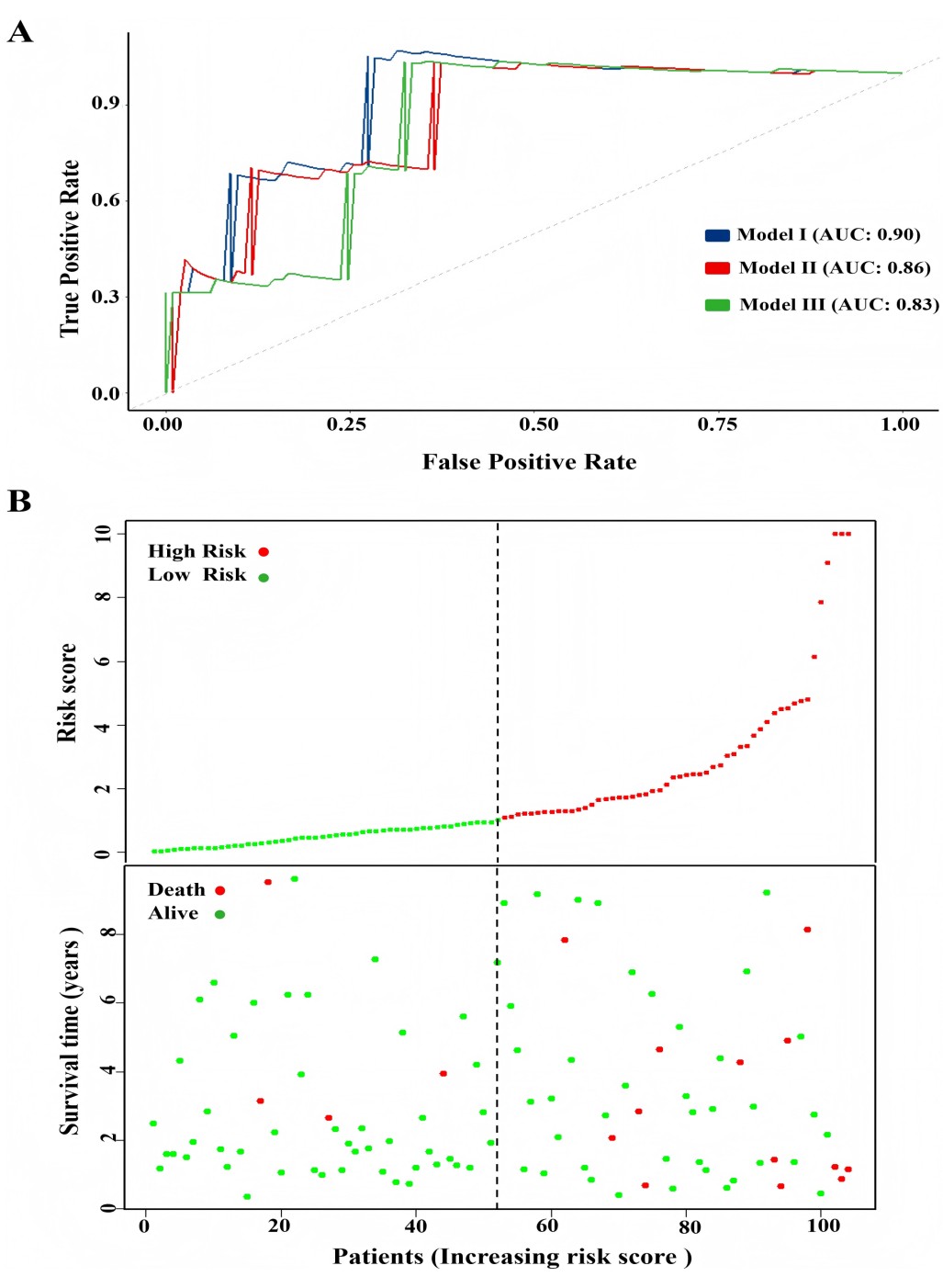

**Figure 6** **Comparison of model performance and risk stratification analysis.** (A) ROC curves for model performance comparison. (B) Risk score distribution and survival status stratification of patients.

Similarly, hsa_circ_0001666 modulated *GFRA1* through hsa-miR-300 and hsa-miR-381-3p, while hsa_circ_0000632 regulated *CA12* through the hsa-miR-520 family (a/b/c/d/e), hsa-miR-302 family (a/b/d/e), and hsa-miR-372/373-3p. Furthermore, cirRNAs such as

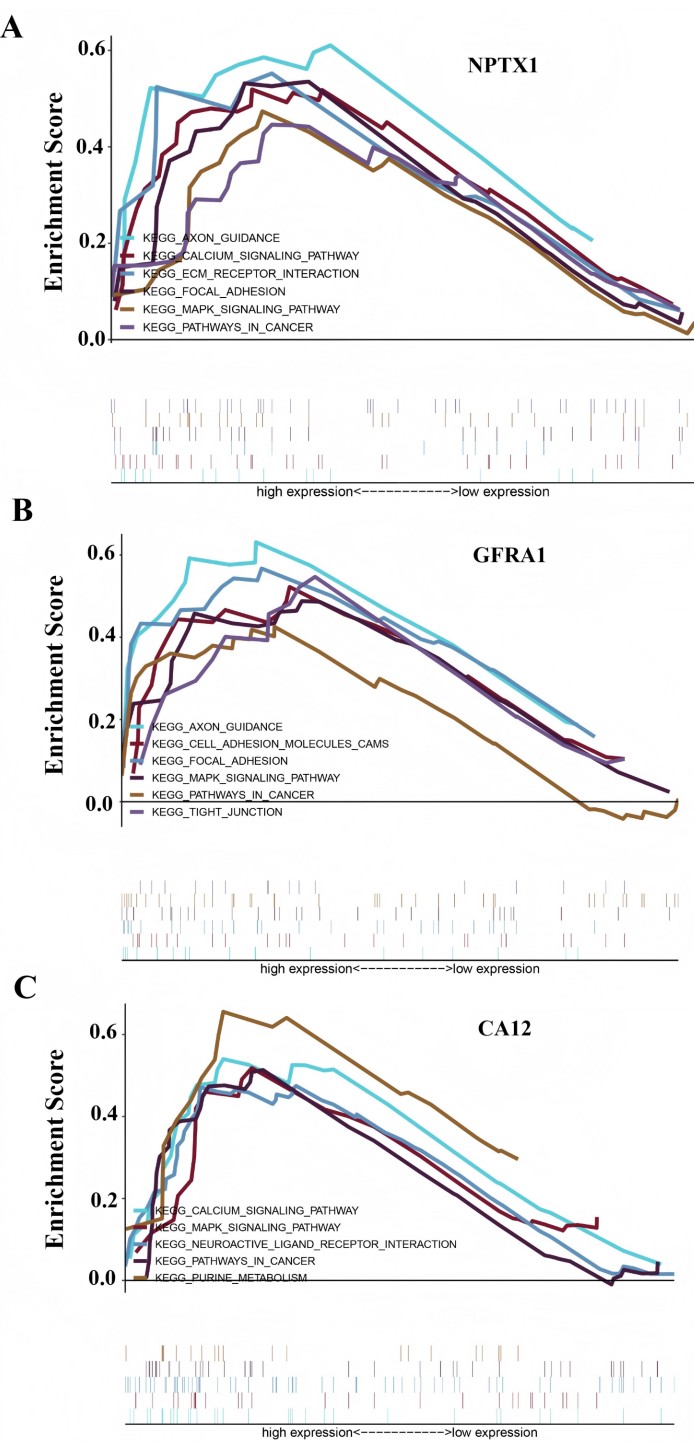

**Figure 7** **GSEA result showing the KEGG pathway of the prognostic model.** (A) The GSEA analysis of the *NPTX1*. (B) The GSEA analysis of the *GFRA1*. (C) The GSEA analysis of *CA12*.

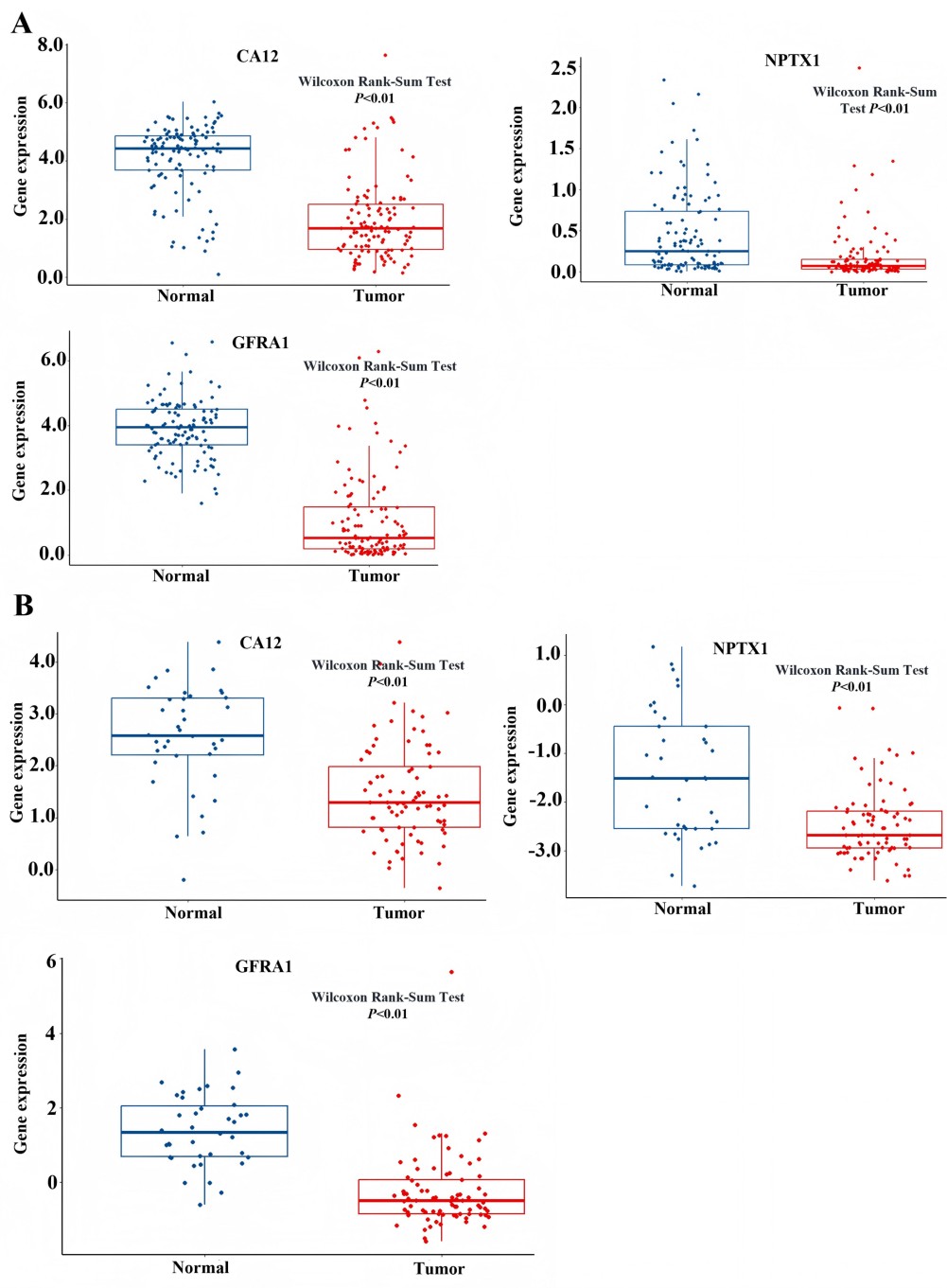

**Figure 8** **Expression level of *NPTX1*, *CA12*, *GFRA1*.** (A) The expression profile of *NPTX1*, *CA12*, *GFRA1* in TCGA dataset. (B) The expression profile of *NPTX1*, *CA12*, *GFRA1* in GEO dataset.

hsa_circ_0005455 and hsa_circ_00000069 targets hsa-miR-135b-5p, thereby co-regulating NPTX1 expression. Our analysis confirmed the significant differential expression of seven key lncRNAs: ADAMTS9-AS1, MAGI2-AS3, ADAMTS9-AS2, DIO3OS, WDFY3-AS2, MEG3, and TPRG1-AS1 (Fig. 10). Among the 22 miRNAs identified in our network,

three miRNAs (miR-139-5p, miR-130a-5p, and miR-135b-5p) demonstrated significant associations with clinical outcomes in TNBC patients (Fig. 11). Notably, validation using the TCGA database provided strong evidence for the clinical relevance of these regulatory networks. These TCGA-based findings substantially support the clinical significance of our identified regulatory network components.

## DISCUSSION

TNBC is one of the most malignant and aggressive breast cancer subtypes, characterized by the absence of ER, PR, and HER2, and it exhibits a higher rate of recurrence and mortality compared to other subtypes (*Bianchini et al., 2016*). Increasing evidences have reported that circRNAs critically influence the modulation of TNBC development (*Mei et al., 2020*). *Xu et al. (2019)* revealed that circTADA2 suppressed the progression and metastasis of TNBC through targeting miR-203a-3p/SOCS3 axis. CircKIF4A modulated the proliferation and migration of TNBC *via* interacting with miR-375 (*Tang et al., 2019*). In *Chen et al. (2018)*, circEPSTI1 was considered a positive predictor for the progression of TNBC, and the knockdown of the circEPSTI1 constrained the TNBC cell proliferation and caused apoptosis (*Chen et al., 2018*). The functions of circRNAs included acting as ceRNA or miRNA sponges, regulating of gene transcription, and functioning as a tumor promoter or suppressor (*Legnini et al., 2017*; *Wang et al., 2018*; *Zhang et al., 2019*). CircRNAs performed the standard biological process in the ceRNA model. *Zhao et al. (2019)* employed bioinformatics mining technology to identify the breast cancer-associated circRNAs through ceRNA analysis. However, most of the ceRNA networks followed the circRNA-miRNA-mRNA model (*Zhao et al., 2019*). Given that lncRNA also competitively targeting the miRNA in the progression of TNBC (*Liu et al., 2019*), we incorporated the DElncRNAs and constructed a lncRNA-miRNA-circRNA-mRNA ceRNA network model of TNBC based on microarray data in this research. Notably, we identified the MAPK signaling pathway as a central regulatory hub, consistent with recent findings demonstrating the importance of ceRNA regulatory networks in TNBC pathogenesis (*Wang et al., 2023*; *Yu et al., 2024*).

It is the first study reporting the profile of the CeRNA (CircRNAs/LncRNAs/miRNAs/mRNAs) network, including both circRNAs and LncRNAs in TNBC. The TNBC-associated ceRNA consisted of 15 circRNAs, 34lncRNAs, 73miRNAs, and 107mRNAs, with the MAPK signal pathway playing a pivotal role in the tumorigenesis process of TNBC. Activation of MAPK signaling has been associated with the evasion and improved clinical prognosis in TNBC (*Loi et al., 2016*). Additionally, we utilized the TCGA dataset as a validation cohort and found that *SH3BGRL2*, *CA12*, *LRP8*, *NAV3*, *GFRA1*, *DCDC2*, *CDC7*, *ABAT*, and *NPTX1* were co-expressed in the TCGA and ceRNA network. The prognostic model based on the nine DEGs had a close relationship with the prognosis of TNBC. The GSEA of the nine DEGs indicated that high expression of *NPTX1*, *GFRA1*, and *CA12* was significantly associated with the MAPK signal pathway. Furthermore, CircGFRA1 had been proved previously to serve as ceRNA in TNBC, modulating GFRA1 expression through sponging miR-34a (*He et al., 2017*). The ceRNA network associated with the MAPK signal

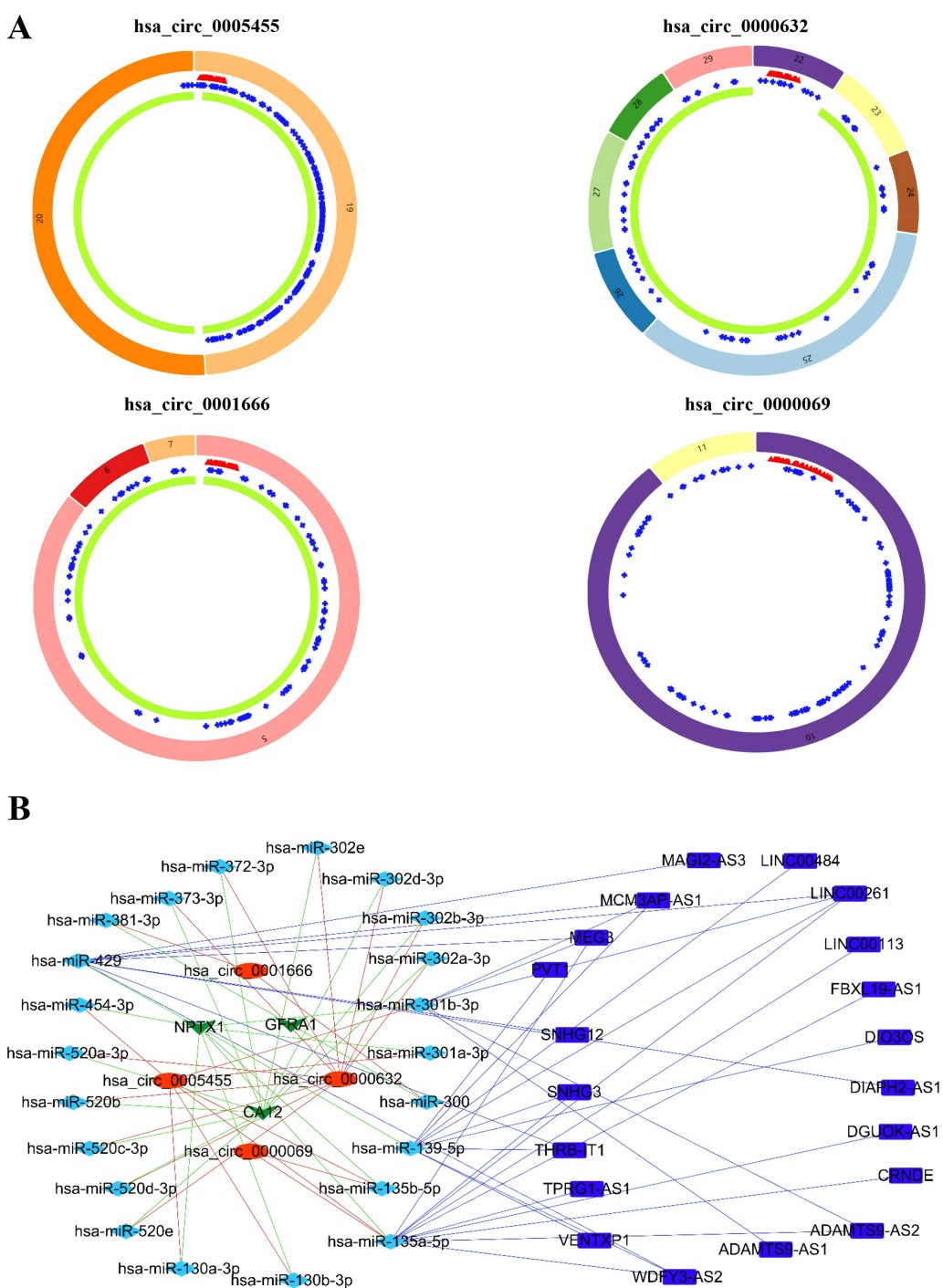

**Figure 9 Identification of the CircRNAs associated with MAPK signal pathway.** (A) The structure pattern of the circRNAs. (B) The CeRNA network associated with MAPK signal pathway.

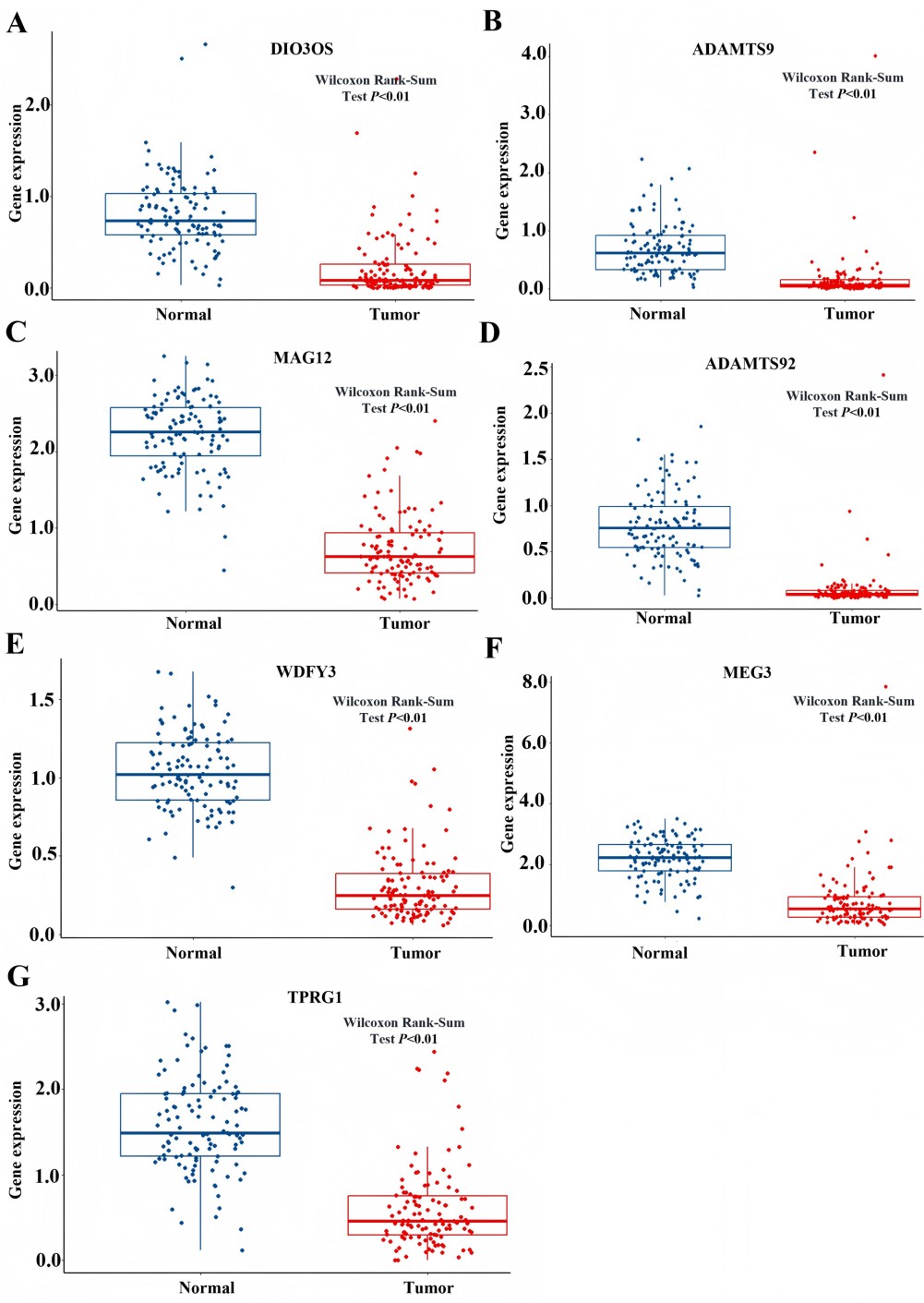

**Figure 10 The expression profile of LncRNAs associated with CeRNA network from TCGA.** (A) Expression of DIO3Os in normal and tumor samples. (B) Expression of ADAMTS9 in normal and tumor samples. (C) Expression of MAG12 in normal and tumor samples. (D) Expression of ADAMTS92 in normal and tumor samples. (E) Expression of WDFY3 in normal and tumor samples. (F) Expression of MEG3 in normal and tumor samples. (G) Expression of TPRG1 in normal and tumor samples.

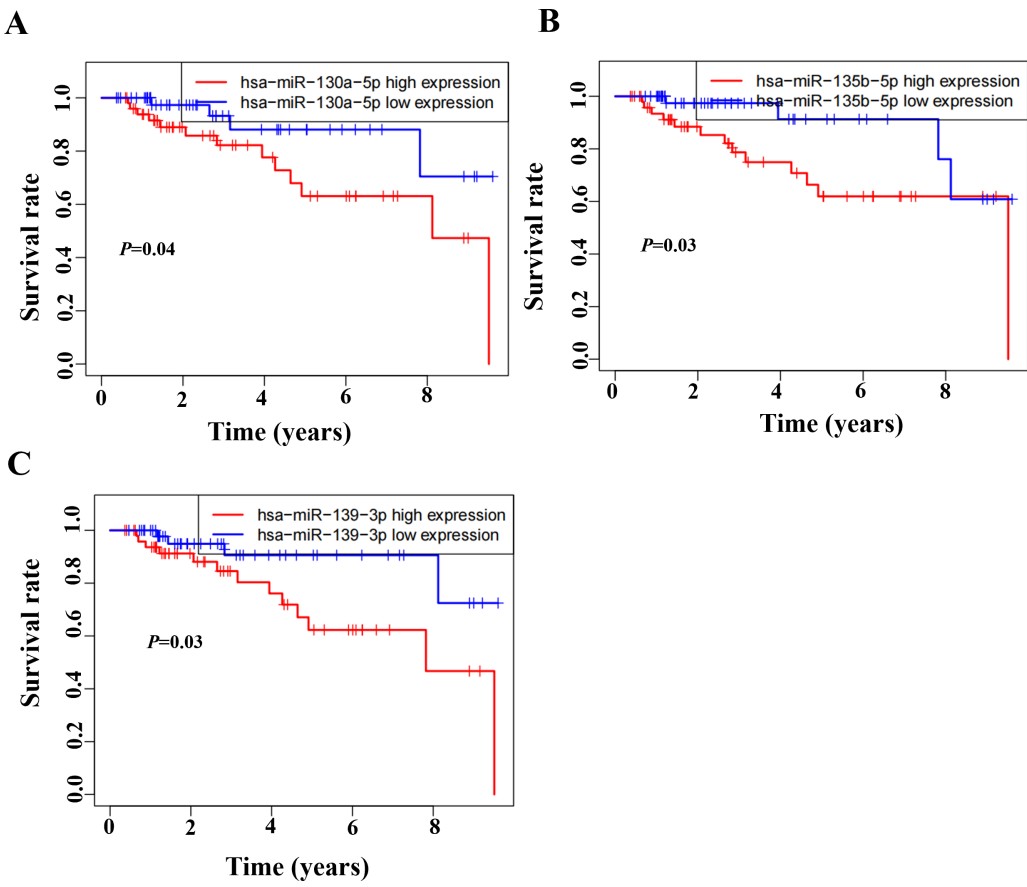

**Figure 11 Survival analysis of prognostic miRNAs associated with the CeRNANetwork TCGA.** (A) Kaplan–Meier survival curve for hsa-miR-130a-5p expression. (B) Kaplan–Meier survival curve for hsa-miR-135b-5p expression. (C) Kaplan–Meier survival curve for hsa-miR-139-3p.

pathway demonstrated that hsa_circ_0005455, hsa_circ_000632, hsa_circ_0001666, and hsa_circ_0000069 were involved in regulating the activity of the *NPTX1*, *GFRA1*, and *CA12*. Further analysis identified seven differentially expressed lncRNAs (ADAMTS9-AS1, MAGI2-AS3, ADAMTS9-AS2, DIO3OS, WDFY3-AS2, MEG3, and TPRG1-AS1) associated with the ceRNA-MAPK signaling pathway that exhibited significant differential expression in TCGA dataset. Notably, decreased expression of miR-139-5p, miR-130a-5p, and miR-135b-5p correlated with improved overall survival in patients with TNBC. Within the ceRNA network, we found that hsa_circ_0005455 and hsa_circ_00000069 potentially co-regulate NPTX1 expression by targeting hsa-miR-135b-5p, with hsa_circ_0005455 and hsa-miR-135b-5p exhibiting the highest connectivity degree in the network (Table S6).

Despite the significant findings of our study, it is important to acknowledge several limitations. First, our analysis primarily relies on bioinformatic approaches and public databases, which may not fully capture the complex biological interactions in TNBC tissue microenvironments. The ceRNA regulatory networks we identified require further experimental validation. Second, the circRNA expression data used in our study was derived

from microarray analysis, which may not detect all circRNA isoforms due to technical limitations. Additionally, while we identified several circRNAs associated with the MAPK pathway, their exact molecular mechanisms and potential protein-coding capabilities require further investigation. Third, although our prognostic model shows promising results, it was developed and validated using retrospective data. External validation using prospective cohorts from multiple institutions would be necessary to establish its clinical utility.

## CONCLUSION

In summary, this study preliminarily indicated that the prognostic model based on the ceRNA network was associated with clinical outcome, and the MAPK signal pathway played a critical role in the progress of TNBC. Importantly, we found that the has_circ_0005455, has_circ_0000632, has_circ_0001666, and has_circ_00000069 were involved in modulating the MAPK signal pathway in the progression of TNBC. The results provided novel insights into the underlying mechanism of TNBC progression and promoted the development of biomarkers in predicting the prognosis of patients with TNBC.

## ACKNOWLEDGEMENTS

We acknowledge the GEO and TCGA databases for providing their platforms and the contributors for uploading these valuable datasets. We also thank all participants involved in the studies included in the current research.

### Funding
The authors received no funding for this work.

### Competing Interests
The authors declare there are no competing interests.

### Author Contributions
- Yimin Zhu conceived and designed the experiments, performed the experiments, analyzed the data, prepared figures and/or tables, and approved the final draft.
- Jiayu Wang conceived and designed the experiments, prepared figures and/or tables, and approved the final draft.
- Binghe Xu conceived and designed the experiments, prepared figures and/or tables, authored or reviewed drafts of the article, and approved the final draft.

### Data Availability
   The raw data are available in the Supplementary Files.
   The RNA sequencing data for all breast cancer samples are available at The Cancer Genome Atlas database: https://cancergenome.nih.gov.

The gene expression profiles are available at NCBI GEO: GSE101123, GSE64790, GSE115275, GSE38959, and GSE53752.

The code is available at Zenodo: Zhu, Y. (2025). Raw data for code. Zenodo. https://doi.org/10.5281/zenodo.14643578.

## Supplemental Information

Supplemental information for this article can be found online at http://dx.doi.org/10.7717/peerj.19063#supplemental-information.

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
