# Peer review of "Development of a prognostic model based on the ceRNA network in Triple-Negative Breast cancer"

_PeerJ, doi:10.7717/peerj.19063_

## Round 0.1 · original submission · Major Revisions

Thank you for your submission. As you can see in the Reviews, Extensive Major Revisions are required for this manuscript to be further considered in PeerJ. Additionally, the revisions below must be included.

In addition to addressing all reviewer concerns, it is also critical that the Methods be presented in a way that is reproducible. This means including critical details (e.g., package versions, parameters, database versions, etc. for ALL software and data sets used), as well as providing the code with a DOI (e.g., through FigShare or Zenodo) to fully reproduce all results in the manuscript (e.g., all figures, tables, and values in the manuscript) as required by the journal.

Additionally:
-a more comprehensive and thorough survey of the current literature should be cited to place this study in the context of both the clinical questions and implications, as well as prior studies.
-The Figures as presented are not acceptable for publication. For example, we cannot read the text on the figures (example: Figure 1, 2, 4, 7, 8, and 9, but please ensure for all)
-The analysis for Figure 5 should also include a PR curve and analysis
-Gene names should be properly italicized
-The tests used to compute all p-values need to be stated with any assumptions; same with multiple hypothesis testing

Thank you.

·

Basic reporting

In the title of paper, I suggest authors to remove index and use model.
Increasing researches. Change to increasing evidences
Use full name for the abbreviations for the first time that they are used
The description of breast cancer and circRNAs should be separated in the introduction of paper.

Experimental design

The experimental part is desirable

Validity of the findings

They are valid

Additional comments

No more comments

·

Basic reporting

Manuscript ID Submission ID 57501v1
This paper is related to reviewing the manuscript titled " Development of prognostic index based on the ceRNA network in triple-negative breast cancer"
To discover possible regulatory relationships, the authors conducted a thorough analysis of differentially expressed circRNAs, mRNAs, and lncRNAs in several GEO datasets. The authors built a competitive endogenous RNA (ceRNA) network based on these data. To confirm our findings, the authors looked at the expression of specific genes in the TCGA TNBC dataset (SH3BGRL2, CA12, NAV3, GFRA1, ABAT, and NPTX1). The authors observed that these genes were differently expressed, and patients with a high-risk score based on these genes had a considerably lower overall survival time. According to authors, the innovative technique provides insight on the fundamental processes of TNBC growth while also providing a viable tool for forecasting patient survival.

Experimental design

Firstly, Although the proposed study is successful in terms of organization, presentation, content and results, major revision given in the following items need to be performed.
1) Provide the major numerical findings and conclusions of the study in the abstract section another than accuracy performance criterion. Also, develop the results section, improve the article to convey the purpose, objectives, method and major findings.
2) Use abbreviations after the first use in the text, in the abstract and throughout the paper.
3) The literature review is quite insufficient in the introduction section. Complete the introduction and literature sections of the article by providing similar studies from the years 2023-2024 and/or new and current studies that will attract the attention of the readers.
4) Explain the concepts of mRNA/lncRNA/circRNAs in TNBC in more detail and verify with numerical expressions.
5) Redraw Figure 8 (Identification of the CircRNAs associated with MAPK signaling pathway) more clearly and explain in detail.

Validity of the findings

.

·

Basic reporting

no comment

Experimental design

no comment

Validity of the findings

This is a pure bioinformatics analysis study, without any wet experiment to validate the findings.

·

Basic reporting

I commend the authors for their extensive dataset, particularly the in-depth biological analyses and the thoughtful integration of clinical data. The manuscript is clearly written and provides valuable insights. The inclusion of additional experimental validation and more detailed clinical data will enhance the strength of this work. Once these revisions are made, I believe the manuscript will be suitable for publication.

Experimental design

The manuscript would benefit from additional biological experimental data to validate the claims made in the text like the ceRNA mentioned.

Validity of the findings

Although the manuscript includes clinical data, it lacks sufficient depth and connection to the experimental findings. Please provide a more detailed description of the patient cohort, including inclusion/exclusion criteria, clinical characteristics, and treatment history. Furthermore, the manuscript would be stronger with statistical analysis of the clinical data, such as Kaplan-Meier survival curves, Cox regression, or other relevant analyses. For instance, in Lines 157-174, the clinical data are presented, but no clear statistical comparisons are made. This diminishes the strength of the argument that the experimental findings correlate with clinical outcomes. It would be beneficial to include these analyses to provide a more compelling, data-driven argument.

Additional comments

While the manuscript is generally well-written, there are areas where the language could be clearer. Some phrases are difficult to understand and may confuse international readers. For example, in Lines 23, 77, 121, and 128, the current phrasing makes comprehension difficult. I recommend that the authors review these sections with a colleague proficient in English or consult a professional editing service. Improved clarity would enhance the manuscript’s accessibility and ensure that the message is communicated effectively to a broad audience. Below are specific suggestions for revision:

Line 23: "The results were not as expected due to variations in the initial setup..." could be rephrased to "The results deviated from our expectations due to inconsistencies in the initial setup..."
Line 128: "This suggests that there may be some potential influence of the variable factors" could be rephrased to "This suggests that variable factors may have influenced the results."

---

## Round 0.2 · accepted · Accept

Thank you for addressing all of the reviewers' comments and the additional issues I identified. This manuscript is now ready for publication.

·

Basic reporting

I recommend accepting this manuscript for publication. The authors have addressed my major concerns regarding of the methodology/the statistical analysis/the discussion of the results.

Experimental design

I recommend accepting this manuscript for publication. The authors have addressed my major concerns regarding of the methodology/the statistical analysis/the discussion of the results.

Validity of the findings

I recommend accepting this manuscript for publication. The authors have addressed my major concerns regarding of the methodology/the statistical analysis/the discussion of the results.

Additional comments

I recommend accepting this manuscript for publication. The authors have addressed my major concerns regarding of the methodology/the statistical analysis/the discussion of the results.

·

Basic reporting

fine

Experimental design

fine

Validity of the findings

fine

Additional comments

I have been satisfied with the revised manuscript.

·

Basic reporting

I have no more comments.

Experimental design

I have no more comments.

Validity of the findings

I have no more comments.

Additional comments

I have no more comments.